# A Novel *DSP* Truncating Variant in a Family with Episodic Myocardial Injury in the Course of Arrhythmogenic Cardiomyopathy—A Possible Role of a Low Penetrance *NLRP3* Variant

**DOI:** 10.3390/diagnostics10110955

**Published:** 2020-11-16

**Authors:** Przemysław Chmielewski, Grażyna T. Truszkowska, Piotr Kukla, Joanna Zakrzewska-Koperska, Mateusz Śpiewak, Małgorzata Stępień-Wojno, Maria Bilińska, Anna Lutyńska, Rafał Płoski, Zofia T. Bilińska

**Affiliations:** 1Unit for Screening Studies in Inherited Cardiovascular Diseases, National Institute of Cardiology, 04-628 Warsaw, Poland; pchmielewski@ikard.pl (P.C.); mstepien@ikard.pl (M.S.-W.); 2Department of Medical Biology, National Institute of Cardiology, 04-628 Warsaw, Poland; gtruszkowska@ikard.pl (G.T.T.); alutynska@ikard.pl (A.L.); 3Department of Cardiology and Internal Diseases, Specialistic Hospital, 38-300 Gorlice, Poland; kukla_piotr@poczta.onet.pl; 41st Department of Arrhythmia, National Institute of Cardiology, 04-628 Warsaw, Poland; jzakrzewska@ikard.pl (J.Z.-K.); mbilinska@ikard.pl (M.B.); 5Magnetic Resonance Unit, National Institute of Cardiology, 04-628 Warsaw, Poland; mspiewak@ikard.pl; 6Department of Medical Genetics, Medical University of Warsaw, 02-106 Warsaw, Poland

**Keywords:** arrhythmogenic cardiomyopathy, desmoplakin, *DSP* gene, *NLRP3* gene

## Abstract

Mono-allelic dominant mutations in the desmoplakin gene (*DSP*) have been linked to known cardiac disorders, such as arrhythmogenic right ventricular cardiomyopathy and dilated cardiomyopathy. During the course of *DSP* cardiomyopathy, episodes of acute myocardial injury may occur. While their mechanisms remain unclear, myocarditis has been postulated as an underlying cause. We report on an adolescent girl with arrhythmogenic biventricular cardiomyopathy and three acute myocarditis-like episodes in whom we found a novel truncating *DSP* variant accompanied by a known low penetrance R490K variant in the *NLRP3*. Upon family screening, other carriers of the *DSP* variant have been identified in whom only mild cardiac abnormalities were found. We hypothesized that the uncommon course of cardiomyopathy in the proband as well as striking discrepancies in the phenotype observed in her family may be explained by the co-existence of her low penetrance genetic autoinflammatory predisposition.

## 1. Introduction

The term “arrhythmogenic cardiomyopathy” (ACM), although poorly defined, refers to a family of diseases that shares structural myocardial abnormalities with ventricular arrhythmia [1]. It has been coined to emphasize, among other things, the overlap between arrhythmogenic right ventricular cardiomyopathy (ARVC) and dilated cardiomyopathy (DCM) when heart failure (HF) caused by either could be linked to desmosomal mutations [2,3]. The diagnosis of ARVC is currently established using the commonly accepted 2010 Task Force criteria [4]. Recent Padua criteria were proposed to modify the diagnostic standards of ARVC, as well as to create missing diagnostic criteria for arrhythmogenic left ventricular cardiomyopathy (ALVC) [5]. They emphasized two main common features of both forms of ACM: fibrosis found on cardiac magnetic resonance (CMR) imaging and the presence of causative mutations. Pathogenic mutations responsible for ACM are found in desmosomal genes encoding: plakophilin 2 (PKP2), desmoplakin (DSP), desmoglein-2 (DSG2), desmocollin-2, junction plakoglobin, which account for the disease in the majority of patients with ARVC [1], and in other genes, of which the most important encode: desmin [1,6], lamin A/C [7,8,9], sodium voltage-gated channel alpha subunit 5 (SCN5A) [10,11], filamin C [12], phospholamban [13], and recently N-cadherin [14].

Mono-allelic dominant mutations in *DSP* have been associated with known cardiac disorders, like classic ARVC [15] and DCM [2,16], as well as sudden cardiac arrest [17] and sudden cardiac death [18]. Bi-allelic recessive *DSP* mutations cause Carvajal syndrome (syndromic form of DCM with palmoplantar keratoderma and woolly hair) [19,20,21] and familial nonsyndromic DCM [22]. Moreover, Smith et al. proposed that the *DSP* cardiomyopathy is a distinct form of ACM, characterized by episodic myocardial injury and left ventricular fibrosis along with a high incidence of ventricular arrhythmias that precede systolic dysfunction [23].

Recently, two case reports linked the onset of *DSP* cardiomyopathy to episodes of myocarditis [24,25]. The inflammation of the myocardium is often present in patients with ACM. Nonetheless, it is not clear whether it is a driving force or only a secondary phenomenon [26]. Differential diagnosis from myocarditis may be challenging. The results of the study by Chelko et al. suggest that the activation of nuclear factor-κB (NF-κB), a transcription factor that plays a key role in the regulation of inflammation, immune response, and apoptosis, may be an important mechanism engaged in ACM development [27,28].

The NF-κB signaling pathway can be activated by cytosolic multiprotein complexes called inflammasomes [29]. The NLR Family Pyrin Domain Containing 3 protein (NLRP3), or cryopyrin, is an important component of the NLRP3 inflammasome. *NLRP3* gain-of-function mutations are the cause of a group of autoinflammatory diseases called cryopyrin-associated periodic syndromes (CAPS) [30], typically characterized by recurrent fever, urticarial rash and arthralgia. Unlike these pathogenic mutations, other low penetrance *NLRP3* variants, also found at low frequencies in control populations, can be associated with atypical CAPS with frequent gastrointestinal symptoms [31]. Inflammation in typical CAPS is driven by the cleavage of caspase 1 and the release of proinflammatory cytokines interleukin-1β (IL-1β) and interleukin-18 (IL-18), another important effect of inflammasome activation. In contrast, disease symptoms in low penetrance *NLRP3* variant carriers are likely to depend on other mechanisms, possibly through the NF-κB pathway [32,33].

The aim of this family case presentation was to characterize ACM due to a novel truncating *DSP* variant and to suggest a link between the episodic myocardial injury and autoinflammatory syndrome predisposition associated with a known low penetrance R490K variant in the *NLRP3* gene.

## 2. Methods

Data from the proband and her relatives were retrospectively collected. They underwent a clinical examination, 12-lead electrocardiography, two-dimensional Doppler echocardiography, blood sampling for genetic testing, and, when justified, 24-hour Holter ECG monitoring and CMR.

DNA was extracted for the purpose of genetic testing from the peripheral blood by phenol extraction. In proband next generation sequencing (NGS) was performed on HiSeq1500 using libraries prepared with TruSight One sequencing panel of 4813 genes (Illumina, San Diego, CA, USA). Bioinformatics analysis was performed as previously described [34]. The detected variants were annotated using Annovar and converted to Microsoft Access format date for final manual analyses. Alignments were viewed with the Integrative Genomics Viewer (IGV). Variants identified in proband were followed up in relatives with Sanger sequencing using a 3500xL or 3130xl Genetic Analyzer (Applied Biosystems, Foster City, CA, USA) and BigDye Terminator v3.1 or v1.1 Cycle Sequencing Kit (Applied Biosystems, Foster City, CA, USA) according to the manufacturer’s instructions. The results were analyzed with Variant Reporter 1.1 Software (Applied Biosystems, Foster City, CA, USA).

The proband and relatives gave written informed consent for DNA analysis and publication of data. 

The genetic testing was performed within the NCBiR ERA-CVD DETECTIN-HF/2/2017 IB.4/II/17 grant, approved on 13 November 2017. The study was approved by the Bioethic Committee of the National Institute of Cardiology, Warsaw. The study conformed to the principles of the Declaration of Helsinki.

## 3. Case Description

The patient’s case history starts with an acute episode which occurred at the age of 15 years, temporarily associated with a wasp sting and a strenuous exercise. She complained of massive chest pain, weakness and fainting and was admitted to the local hospital. Her troponin T level was elevated at max. 2280 ng/L (normal range < 14 ng/L), whilst the N-terminal prohormone B-type natriuretic peptide (NT-proBNP) and C-reactive protein (CRP) levels were normal. The ECG abnormalities included low QRS voltages and frequent premature ventricular contractions (PVCs). On the echocardiogram, the left ventricle (LV) was mildly dilated and hypokinetic with left ventricular ejection fraction (LVEF) of 47%, no right ventricle (RV) abnormalities were detected.

The next episode occurred two years later and was associated with symptoms of upper respiratory tract infection. It ran with massive chest pain, palpitations and dyspnea on mild effort. The troponin T level went up to max. 522 ng/mL. The ECG showed new negative T waves in the right precordial leads. On the echocardiogram, LV was mildly dilated (55 mm), lateral wall and septum hypokineses were found with LVEF 47%, and RV was also dilated with areas of segmental hypokinesis. The N-terminal prohormone B-type natriuretic peptide (NT-proBNP) level was elevated with max. value of 478 pg/mL, whereas the CRP level was normal. 

She received typical HF treatment (metoprolol, ramipril, spironolactone) and within two weeks after the onset of each episode her troponin T level (Figure 1) and LVEF returned to normal but, still, she complained of dyspnea on effort and palpitations. During the follow-up after the second incident, the NT-proBNP level remained elevated (455 pg/mL) and frequent multifocal PVCs, as well as episodes of idioventricular rhythm and salvoes of polymorphic non-sustained ventricular tachycardia (nsVT) were recorded on Holter monitoring. 

At the age of 18 years, she was referred to the National Institute of Cardiology, Warsaw, and underwent renewed full diagnostic work-up, including genetic evaluation. The findings were consistent primarily with the clinical picture of ALVC, namely: (1) low QRS voltage in the limb and precordial leads on ECG; in addition to the terminal activation duration of QRS >55 ms, QRS fragmentations in leads II, III and aVF, multifocal PVCs originating in both ventricles but no negative T-waves in right precordial leads (Figure 2); (2) frequent multifocal PVCs on Holter monitoring; (3) mildly dilated left ventricle with lateral wall and septum hypokineses, and an LVEF of 40%; and (4) moderate sub-epicardial and mid-wall areas of late gadolinium enhancement (LGE) with a ring-like pattern demonstrated on CMR (Figure 3). She also fulfilled the 2010 Task Force Criteria for the diagnosis of ARVC, considering increased RV volume (121 mL/m^2^), systolic dysfunction (RV ejection fraction of 35%) with areas of segmental akinesis, documented on echo and CMR, along with the depolarization abnormalities on ECG and frequent ventricular arrhythmia.

Regarding systemic symptoms, the patient reported strong recurrent abdominal pain, relieved by vomiting, but she negated other symptoms such as fever, urticarial rash or joint pain.

In the proband whole exome sequencing was performed (Figure 4). The sequencing run for the patient sample achieved 25,784,450 reads with a mean coverage of 35.64. Above 76.7 and 94% of the target region was covered a minimum 20 and 10 times, respectively. We identified heterozygous variants: (1) *DSP* NM_004415.4:c.3737dupA (p.Asn1246LysfsTer7), a predicted pathogenic variant in the *DSP* gene; (2) *PKP2* NM_004572.3:c.2636T>C (p.Leu879Pro), a variant of unknown significance in the plakophilin-2 (*PKP2*) gene, and (3) *NLRP3* NM_004895.4:c.1469G>A (p.Arg490Lys), a known low penetrance variant in the *NLRP3* gene. 

At the age of 19 years, she had another episode of long-lasting chest pain, accompanied by new ECG changes (negative T waves in I, aVL) and an elevated troponin T level, which peaked from normal to 4010 ng/L and dropped quickly afterwards. No new wall-motion abnormalities were found on the echocardiography. CRP levels were low–normal. NT-proBNP, elevated already before the episode (734 pg/mL), was raised to 1164 pg/mL and persisted as elevated 12 months after the incident (1349 pg/mL).

Subsequent CMR showed a similar degree of systolic dysfunction of both ventricles (LVEF 34%, RVEF 35%) to the previous investigation but the significant progression of fibrosis within LV (Figure 3). The patient received an implantable cardioverter–defibrillator at the age of 21 years.

Currently, at the age of 24 years, the patient is in New York Heart Association functional class II, with stably reduced LVEF of approx. 40% and persistent, mildly symptomatic ventricular ectopy. She receives typical HF treatment (metoprolol of 100 mg/day, torasemide of 5 mg/day, ramipril of 5 mg/day, eplerenone of 25 mg/day). Systemic symptoms receded.

The patient’s family history was unremarkable, except for an episode of frequent ventricular arrhythmia in her brother. Her parents and siblings underwent clinical evaluation (ECG, echo) and genetic screening for the variants identified in the proband (results are shown in Figure 5). All clinical studies were performed with investigators blinded to the results of genetic testing.

The proband’s only brother (18 years old at the genetic inquest) turned out to be the only carrier of all three identified variants. He had a history of asymptomatic arrhythmia, detected in 24-hour Holter monitoring at the age of 17 and consisting of 9215 PVCs of right bundle branch block and left posterior fascicular block morphology which resolved without specific treatment. After the episode, his 12-lead ECG, echocardiogram, CMR, serum biomarkers and repeated Holter monitoring were normal. We found no substantial differences in his lifestyle (incl. strenuous exercise) in comparison to the proband that could account for phenotypic disparities.

In her father, a 59-year-old man with comorbidities (hypertension, obesity, metabolic syndrome) and the *DSP*-variant carrier, mild left ventricular systolic dysfunction was found on the echocardiogram (LVEF 46%), and infrequent PVCs and an episode of polymorphic nsVT on Holter recordings. The CMR, performed two years later, showed LV hypertrabeculation, the improvement of LV systolic function (LVEF 67%), no abnormal findings within RV and merely slight nonspecific LGE in junction points of the RV in the septum. The remaining siblings and their mother were asymptomatic and without signs of cardiac involvement, including two carriers of the *PKP2* variant.

## 4. Discussion

### 4.1. DSP—Novel Variant

The NM_004415.4:c.3737dupA variant in the *DSP* gene has not been reported to our knowledge. This variant causes a frameshift starting at codon Asparagine 1246, changing it to Lysine and creating a premature stop codon at position 7 of the new reading frame, denoted as p. Asn1246LysfsTer7. Loss-of-function variants in *DSP* are known to be pathogenic and are an established disease mechanism in autosomal dominant ARVC and autosomal recessive Carvajal syndrome [19]. This variant is located in exon 23 of the *DSP* gene, which undergoes alternative splicing to produce three isoforms: a short (DSPII), an intermediate (DSPIa), and a long (DSPI) form. The c.3737dupA variant impacts the rod domain of all three isoforms. The long DSPI transcript is the predominant isoform in cardiac tissue and multiple loss-of-function variants affecting this isoform have been identified in individuals with ARVC and/or DCM [35,36].

The NM_004572.3:c.2636T>C (p.Leu879Pro) variant in the *PKP2* gene is classified according to American College of Medical Genetics criteria as a variant of uncertain significance [37] and we cannot provide any evidence to confirm its pathogenicity.

### 4.2. Phenotypic Differences in the DSP Variant’s Carriers within the Family

In this report, we present striking discrepancies in the phenotype of the *DSP* cardiomyopathy. The most severe phenotype was present in an adolescent, who suffered from symptoms of acute myocardial injury thrice. The first episode followed a wasp sting and strenuous physical exertion (walking pilgrimage), lasting several days, the second one accompanied a cold, the last happened without any apparent cause. All episodes were associated with elevation of cardiac biomarkers, however, with normal CRP, and led to significant biventricular myocardial dysfunction. In contrast, her brother, a carrier of all three variants (*DSP*, *PKP2* and *NLPR3*), had one unclear episode of ventricular arrhythmia that disappeared without sequels. Their father, who had the *DSP* variant alone, had relatively mild cardiac abnormalities, detected only on the family screening.

Given the variable clinical manifestations in this family, we had to consider other pathogenic factors in addition to the *DSP* mutation. The possible explanation may be the effect of the low penetrance p. Arg490Lys variant in the *NLRP3* gene, identified in the proband. Pathogenic heterozygous gain-of-function *NLRP3* mutations cause the excessive release of IL-1β and systemic inflammation, and result in full blown CAPS. In contrast, low penetrance *NLRP3* variants including Q703K, V198M and R490K (listed also as R488K according to previous transcripts) are present at low frequencies in control populations [31]. The p.R490K variant is located in exon 3 of the *NLRP3* gene, where most reported pathogenic variants cluster. It was initially described as the cause of CAPS [38,39,40] and later reported as showing a reduced clinical penetrance [41,42]. Interestingly, Haverkamp et al. found that peripheral blood mononuclear cells from healthy *NLRP3* p.R490K variant carriers had higher stimulation indices of IL-1β and INF-gamma than population controls. Furthermore, Rae et al. [41] identified the *NLRP3* p.R490K variant in the context of another pathogenic *GATA2* variant proposing that carrying both variants may result in a blended phenotype of 2 distinct monogenic diseases. 

Kuemmerle-Deschner et al. showed that symptomatic individuals with the R490K and other low penetrance *NLRP3* variants have a distinct clinical phenotype [31], with dominant gastrointestinal symptoms such as abdominal pain, diarrhea, constipation, vomiting and heartburn, which is compatible with our proband’s complaints. The mode of action of low penetrance *NLRP3* variants remains unclear [32,33]. Disease symptoms in these subjects seem independent of the cleavage of caspase 1 and the release of proinflammatory cytokines and are likely to depend on other mechanisms [32,33]. Therefore, the demonstration of functional abnormalities may be challenging, even in the affected individuals. The CRP level was normal in 64–67% of symptomatic patients [31,43], as in our proband. In atypical CAPS, interleukin-6 and serum amyloid A serum concentrations were elevated in only 17 and 36–67%, respectively [31,32]. IL-1β serum measurements do not distinguish even between typical CAPS patients and healthy controls, possibly due to its prompt neutralization and low and variable level [44]. Good candidates for disease indicators in atypical CAPS could be tumor necrosis factor and S100 calcium-binding protein A12, as their circulating levels were elevated in 90 and 100% of cases, accordingly [31]. Unfortunately, we are not able to provide data on these markers in our family. Furthermore, the results of two studies showed no significant differences between low penetrance *NLRP3* variant carriers and wild-type controls in such inflammasome in vitro assays, as caspase 1 activity and cleavage, as well as IL-1β and IL-18 release [31,32].

It should be noted that in the studies by both Aksentijevich et al. [45] and Haverkamp et al. [42], unaffected family members were also found to harbor the p.R490K mutation, indicating that it may be associated with reduced or absent clinical features, as also apparent in the remaining carriers of the *NLRP3* variant from our family. 

In their recent paper, Smith et al. stated that *DSP* cardiomyopathy is a distinct form of ACM characterized by episodic myocardial injury [23]. Acute episodes occurred in 16/107 (15%) patients, and the authors did not mention whether they encountered cases with recurrent episodes. In this landmark study, however, only *DSP* variants were evaluated, collected from six tertiary referral centers. There were no differences in the penetrance of *DSP* cardiomyopathy attributable to sex or exercise burden.

Interestingly, in a family similar to this one, with recurrent myocarditis in two sibs with a *DSP* truncating variant inherited from a parent with a possibly milder phenotype, was reported recently [24]. While the authors did not find functional immunological abnormalities, genetic testing was limited to cardiomyopathy genes raising a possibility that genetic predisposition to immune dysfunction was not detected [24].

We supposed the differences in severity of *DSP* cardiomyopathy between the proband and her mildly affected father might be explained by the concomitant *NLRP3* variant. The lack of symptoms in her younger brother, who carried all identified variants but had no signs of systemic inflammatory disorder, may be explained by his young age, incomplete penetrance of both the *DSP* variant and, particularly, the low penetrance *NLRP3* variant.

## 5. Study Limitations

The patient and her family came to our center from a remote region of Poland for evaluation in a stable phase of the disease. Our report used retrospectively collected data which contained no information on the cytokine or other inflammatory marker serum levels except for CRP. No heart biopsy samples were available for testing, either. Thus, the role of the low penetrance *NLRP3* variant in the pathogenesis of ACM in our case remains a hypothesis and should be considered with alternative options, i.e., the episodic course of ACM attributable solely to *DSP* mutations.

## 6. Conclusions

In summary, we presented a novel pathogenic variant in *DSP* cardiomyopathy. Our case shows that the progression of *DSP* cardiomyopathy follows acute episodes of myocardial injury, which in our family might be related to low penetrance genetic autoinflammatory predisposition. Testing inflammatory genes in ACM patients might be considered in cases with a more severe phenotype or episodic acute myocardial injury.

## Figures and Tables

**Figure 1 diagnostics-10-00955-f001:**
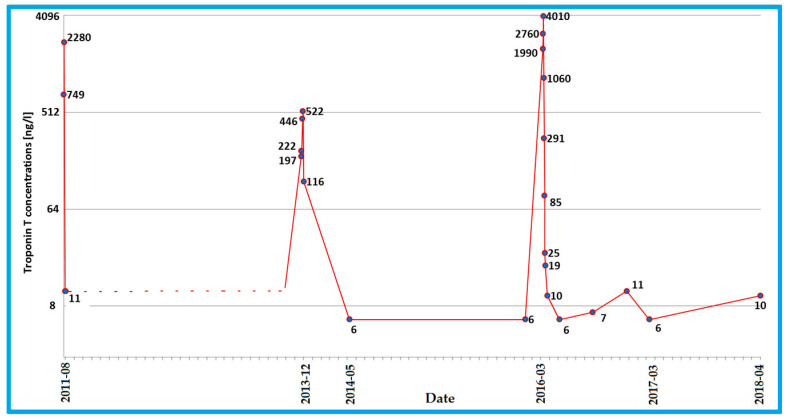
Changes in the troponin T concentration, reflecting the course of the *DSP* cardiomyopathy with recurrent acute myocardial injury. A dotted line was used where no measurements were available to recreate the course of the variable.

**Figure 2 diagnostics-10-00955-f002:**
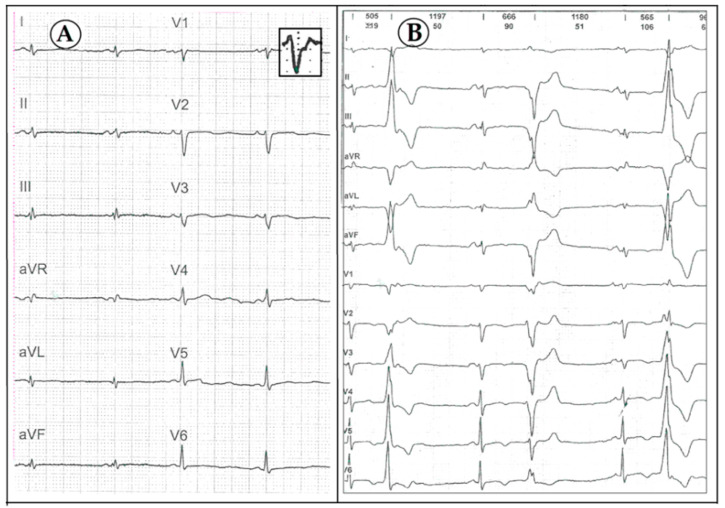
12-lead electrocardiogram recordings in the proband with desmoplakin cardiomyopathy. (**A**) Abnormalities of QRS complex: low voltage, prolonged terminal activation duration in lead V1 (see in the magnified box), fragmentations in leads II, III and aVF; (**B**) multifocal premature ventricular contractions: (1) left bundle branch block (LBBB)-like morphology (QRS transition zone V3) with inferior axis suggests the beginning of activation in the right ventricular outflow tract, (2) LBBB-like morphology (QRS transition zone in V6) with superior axis suggests the origin from the free wall of the right ventricle, periapically; (3) nonspecific intraventricular conduction delay-like morphology (QRS transition zone in V1–2) with inferior axis suggests the beginning of activation in the left ventricular outflow tract.

**Figure 3 diagnostics-10-00955-f003:**
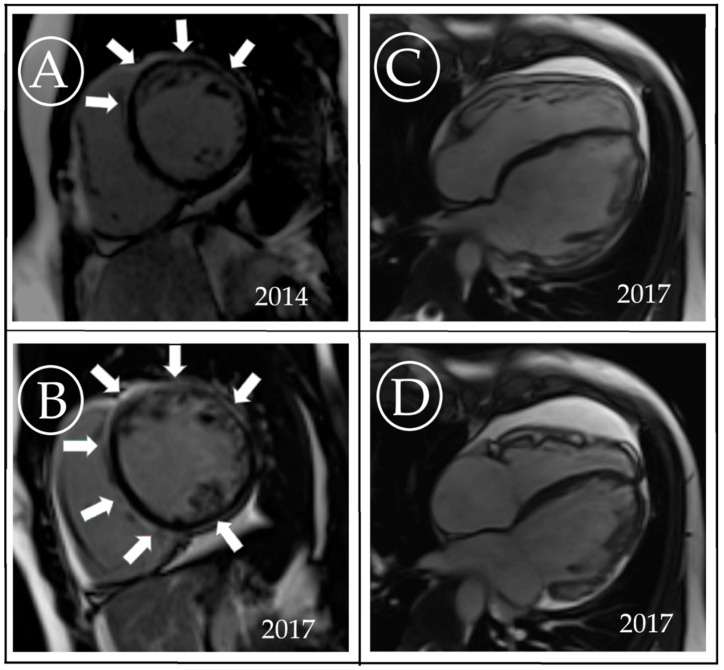
Cardiac magnetic resonance images consistent with the diagnosis of arrhythmogenic cardiomyopathy. (**A**) Short-axis slice demonstrating moderate sub-epicardial and mid-wall areas of late gadolinium enhancement with a ring-like pattern (arrows); (**B**) the progression of ring-like pattern fibrosis (arrows) in comparison to the study performed 3 years earlier; (**C**) steady-state free precession 4-chamber view in the end-diastole; (**D**) steady-state free precession 4-chamber view in end-systole. Areas of bulging of right ventricular wall are seen.

**Figure 4 diagnostics-10-00955-f004:**
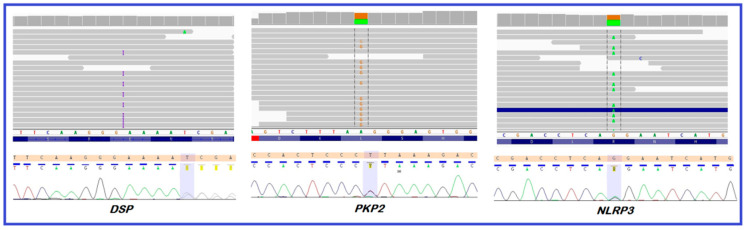
Integrated Genomic Viewer (IGV) views and electropherograms of the identified heterozygous variants: *DSP* NM_004415.4:c.3737dupA (p.Asn1246LysfsTer7), *PKP2* NM_004572.3:c.2636T>C (p.Leu879Pro), and *NLRP3* NM_004895.4:c.1469G>A (p.Arg490Lys).

**Figure 5 diagnostics-10-00955-f005:**
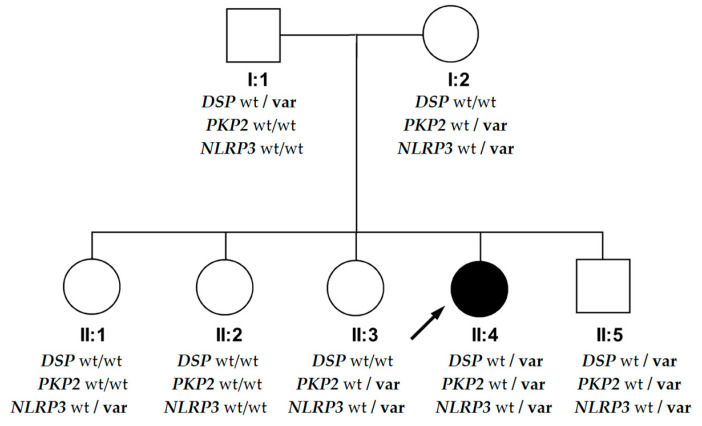
The results of genetic screening for the identified variants, shown on a pedigree. Squares represent males and circles represent females. The arrow denotes the proband. The solid black symbol denotes arrhythmogenic cardiomyopathy. Open symbols denote unaffected individuals. Legend: wt—wild type, var—heterozygous variants: *DSP*—NM_004415.4:c.3737dupA (p.Asn1246LysfsTer7) variant in the desmoplakin gene, *PKP2*—NM_004572.3:c.2636T>C (p.Leu879Pro) variant in the plakophilin-2 gene, *NRLP3*—NM_004895.4:c.1469G>A (p.Arg490Lys) variant in the NLR Family Pyrin Domain Containing 3 gene.

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
