# Peer review of "A Novel DSP Truncating Variant in a Family with Episodic Myocardial Injury in the Course of Arrhythmogenic Cardiomyopathy—A Possible Role of a Low Penetrance NLRP3 Variant"

_diagnostics, 2020, doi:10.3390/diagnostics10110955_

Round 1

Reviewer 1 Report

Heretofore, it was believed that inflammation was a secondary phenomenon in arrhythmogenic cardiomyopathy (ACM); a consequence of cell death. Recent studies suggest that it may be a driving force of the disease. This is the first case report to present a variant in an inflammasome gene in an ACM patient with a desmoplakin mutation. Inflammatory genes should be sequenced in ACM patients for mutations that may either be pathogenic or additive to the disease phenotype. Please see my comments below:

Major

The introduction is poorly written and difficult for the reader to follow unless he/she is an expert in ACM. Please revise the writing style of the manuscript.

Since both the proband and her brother are carriers of the same 3 variants – one would expect that they would show a similar phenotype unless their environment is very different. Was the proband an athlete while the brother had a more sedentary lifestyle?

The authors state that the desmoplakin variant could either act in a dominant negative fashion or through haploinsufficiency. Is it possible for them to establish the mode of action? Is any heart biopsy sample available from the ICD implantation that could be used for a Western blot? Alternatively, a different tissue sample may be used as well such as a skin biopsy or even cells from the buccal epithelium.

First of all – the authors need to refer to a ‘mechanistic link’ between the inflammasome and ACM. The recent paper by Chelko et al in Circulation (2019) shows that the NFκB pathway is in fact a driving pathway in the ACM pathogenesis. Secondly, the authors must try and provide some evidence that this NLRP3 variant may in fact be playing a role in the pathogenesis of the disease. If heart tissue is available, it might be worth immunostaining it for this inflammasome gene as well as for downstream cytokines such as IL1β.

The authors might want to highlight why this case report is important; not because it is yet another DSP variant in an ACM patient but because inflammatory genes should be sequenced in ACM patients that either have no mutations in known genes or a more severe phenotype.

The authors should also refer to the limitations of their study; at least state that they cannot tell what this NLRP3 variant is doing unless in vitro or in vivo animal studies are performed.

Minor

In the introduction, the genes underlying ACM pathogenesis need to be spelled out.

Also, mutations in N-cadherin have been found to underlie cases of ACM as well.

There are several case reports of recessive DSP mutations or compound heterozygotes showing a worse ACM/DCM phenotype; not just the two referenced by the authors.

It might be beneficial for the authors to explain what the NFκB pathway is. It is a master regulator of inflammation but perhaps readers in the cardiology community may not be aware of it.

Since the authors refer to the 2010 Task Force criteria in their case report description it might be useful to include this in their introduction as well along with an appropriate reference.

The authors should spell out CAPS

Author Response

Open Review #1

English language and style

( ) Extensive editing of English language and style required
(x) Moderate English changes required
( ) English language and style are fine/minor spell check required
( ) I don't feel qualified to judge about the English language and style

Yes

Can be improved

Must be improved

Not applicable

Does the introduction provide sufficient background and include all relevant references?

( )

( )

(x)

( )

Is the research design appropriate?

( )

(x)

( )

( )

Are the methods adequately described?

( )

(x)

( )

( )

Are the results clearly presented?

( )

( )

(x)

( )

Are the conclusions supported by the results?

( )

( )

(x)

( )

Comments and Suggestions for Authors

Heretofore, it was believed that inflammation was a secondary phenomenon in arrhythmogenic cardiomyopathy (ACM); a consequence of cell death. Recent studies suggest that it may be a driving force of the disease. This is the first case report to present a variant in an inflammasome gene in an ACM patient with a desmoplakin mutation. Inflammatory genes should be sequenced in ACM patients for mutations that may either be pathogenic or additive to the disease phenotype. Please see my comments below:

We are grateful to the Reviewer for the insightful review and numerous valuable comments. We responded to all questions and modified our manuscript accordingly.

Major

The introduction is poorly written and difficult for the reader to follow unless he/she is an expert in ACM. Please revise the writing style of the manuscript.

We propose a rewritten version of introduction which, hopefully, will be easier to follow.

Since both the proband and her brother are carriers of the same 3 variants – one would expect that they would show a similar phenotype unless their environment is very different. Was the proband an athlete while the brother had a more sedentary lifestyle?

There were no substantial differences in the lifestyle (incl. strenuous exercise) between the proband and her brother. Anyway, Smith et al. found no differences in penetrance of DSP cardiomyopathy attributable to sex or exercise burden. We suppose the asymptomatic phenotype found in the brother may be associated with his young age, incomplete penetrance of the DSP variant and, particularly, the low-penetrant NLRP3 variant – he had no signs of systemic inflammatory disorder.

We added appropriate formulations to the manuscript lines: 237-9 (case description), 322, 330-332 (discussion).

The authors state that the desmoplakin variant could either act in a dominant negative fashion or through haploinsufficiency. Is it possible for them to establish the mode of action? Is any heart biopsy sample available from the ICD implantation that could be used for a Western blot? Alternatively, a different tissue sample may be used as well such as a skin biopsy or even cells from the buccal epithelium.

Unfortunately, a heart biopsy sample is not available, and due to technical and time problems we are not able to perform Western Blot to establish mode of action of DSP variant. The quoted sentence refers to possible mechanisms involved generally in diseases caused by mono-allelic dominant mutations but is not essential for our report. Since we cannot provide evidence which mechanism is responsible for the disease in our case, we removed it from the manuscript. (line 252)

First of all – the authors need to refer to a ‘mechanistic link’ between the inflammasome and ACM. The recent paper by Chelko et al in Circulation (2019) shows that the NFκB pathway is in fact a driving pathway in the ACM pathogenesis.

Thank you for highlighting this. We consider it valuable and included it in the introduction (line 63) with appropriate reference.

“The results of the study by Chelko et al. suggest that activation of nuclear factor-κB (NF-κB), the master regulator of inflammation, immune response, and apoptosis, may be an important mechanism engaged in ACM development.”

Secondly, the authors must try and provide some evidence that this NLRP3 variant may in fact be playing a role in the pathogenesis of the disease.

Unfortunately, we cannot provide evidence of the actual role of the NLRP3 variant – not only because of retrospective character of our report, but also due to failure to demonstrate  consistent differences in laboratory parameters between low-penetrance NLRP3 variant carriers and healthy controls.

Following your suggestion, we carried out a thorough analysis of available literature and found no certain determinants of the affected status of low-penetrance NLRP3 variant carriers.

We elaborate on these issues extensively in the discussion (lines 297-313)

“The mode of action of low penetrance NLRP3 variants remains unclear 29, 30. Disease symptoms in these subjects seem independent of cleavage of caspase 1 and release of proinflammatory cytokines and are likely to depend on other mechanisms 29, 30. Therefore, demonstration of functional abnormalities may be challenging, even in the affected individuals. CRP level was normal in 64-67% of symptomatic patients 28, 39, as in our proband. In atypical CAPS, interleukin-6 and serum amyloid A serum concentrations were elevated in only 17% and 36-67%, respectively28, 29. IL-1β serum measurements do not distinguish even between typical CAPS patients and healthy controls, possibly due to its prompt neutralisation and low and variable  level40. Good candidates for disease indicators in atypical CAPS could be tumor necrosis factor and S100 calcium-binding protein A12, as their circulating levels were elevated in 90% and 100% of cases, accordingly 28. Unfortunately, we are not able to provide data on these markers in our family. Furthermore, the results of two studies showed no significant differences between low-penetrance NLRP3 variant carriers and wild-type controls in such in-vitro inflammasome assays, as caspase 1 activity and cleavage, as well as IL-1β and IL-18 release28, 29.”

and also added the following text to the study limitations.

“Thus, the role of the low-penetrant NLRP3 variant in the pathogenesis of ACM in our case remains a hypothesis and should be considered with alternative options, i.e. episodic course of ACM attributable solely to DSP mutations.”

If heart tissue is available, it might be worth immunostaining it for this inflammasome gene as well as for downstream cytokines such as IL1β.

Unfortunately, heart biopsy is not available, and we do not even have serum measurements of cytokines or other inflammatory markers. On the other hand, the mode of action of low penetrance NLRP3 variants remains unclear but it seems independent of cleavage of caspase 1 and release of proinflammatory cytokines. – These issues are discussed in lines 297-305.

The authors might want to highlight why this case report is important; not because it is yet another DSP variant in an ACM patient but because inflammatory genes should be sequenced in ACM patients that either have no mutations in known genes or a more severe phenotype.

Thank you for highlighting this. We included appropriate text in the conclusions.

“Testing inflammatory genes in ACM patients might be considered in cases with more severe phenotype or episodic acute myocardial injury.”

The authors should also refer to the limitations of their study; at least state that they cannot tell what this NLRP3 variant is doing unless in vitro or in vivo animal studies are performed.

We included appropriate statements  in the discussion and study limitations section.

Minor

In the introduction, the genes underlying ACM pathogenesis need to be spelled out.

Thank you for highlighting this. We corrected the manuscript accordingly.

Also, mutations in N-cadherin have been found to underlie cases of ACM as well.

There are numerous genes associated with ACM phenotype and we listed only some most prominent examples; other include RYR2, TGFB3, CTNNA3, PLN, TMEM43. N-cadherin is yet another example – we added this to the manuscript with appropriate citation.

There are several case reports of recessive DSP mutations or compound heterozygotes showing a worse ACM/DCM phenotype; not just the two referenced by the authors.

We added additional references  (20-21)

It might be beneficial for the authors to explain what the NFκB pathway is. It is a master regulator of inflammation but perhaps readers in the cardiology community may not be aware of it.

Thank you for highlighting this. We included an appropriate passage  in the Introduction section (lines 63-65).

“activation of nuclear factor-κB (NF-κB), the master regulator of inflammation, immune response, and apoptosis, may be an important mechanism engaged in ACM development”

Since the authors refer to the 2010 Task Force criteria in their case report description it might be useful to include this in their introduction as well along with an appropriate reference.

We added appropriate reference in the introduction.

The authors should spell out CAPS

CAPS is spelled out in the introduction (line 68).

Reviewer 2 Report

The authors decribe a case report of a carrier of a novel truncating desmoplakin variant accompanied to a known low-penetrant variant in the NLRP3 gene.

A number of key issues has to be solved.

I recommend that this manuscript will be restructured according to the following suggestions.

The introduction must be improved by exploring in depth the following issues:

  • the authors only refer to the Padua criteria but in the text they state “She also fulfilled the 2010 Task Force Criteria for the diagnosis of ARVC” (lines 106-107). Therefore, it is necessary to mention and comment the 2010 Task Force Criteria for the diagnosis of ARVC.
  • the authors only refer to the ACM overlap with DCM, which is ok, but does not seem to be related to the described case. According to the hypothesis of an “autoinflammatory syndrome predisposition” and the “acute myocarditis-like episodes”, it is essential to discuss the possible overlap between ACM and myocarditis.
  • the description of NLRP3 function is unclear and needs to be expanded as well as the impairement caused by the low-penetrant R490K variant during the autoinflammatory syndrome.

The authors state that "NLRP3 mutations cause excessive release of interleukin 1β and systemic inflammation", but no signs of systemic inflammation (normal levels of C-reactive protein) has been found in the proband. The authors only describe troponin elevation, which may be common in myocarditis but is not specific to myocarditis. I suggest to test interleukin 1β concentration, or that of other inflammatory cytokines in the plasma of the family members.

The fact that the proband’s brother is carrier of all three identified variants and only has a history of asymptomatic arrhythmias, works against the hypothesis. The authors should better explain their view concerning the differential clinical manifestations and the role of physical exercise as possible trigger. It would be important to test other differences (even not genetic) between the two siblings.

Please comment in the discussion the paper: Desmoplakin Cardiomyopathy, a Fibrotic and Inflammatory Form of Cardiomyopathy Distinct From Typical Dilated or Arrhythmogenic Right Ventricular Cardiomyopathy. Circulation 2020;141:1872-1884. It is likely that the inflammatory and episodic nature of the disease of the proband is typical of DSP driven cardiomyopathy. This should be taken into account as a hypothesis alternative to the role of NLRP3 variant.

The method description section is completely missing

Please clarify paragraph 3.3.

Based on the absence of validation studies it is necessary to provide a study limitations section

Please make sure that genes are in Italic (also in the figures)

Normally DSP+ means the wt allele of DSP, while in fig.5 means presence of the mutation. Please modify the figure to make it easily readable.

Please define in Figure 1 legend the dotted line

Author Response

Open Review #2

English language and style

( ) Extensive editing of English language and style required
( ) Moderate English changes required
(x) English language and style are fine/minor spell check required
( ) I don't feel qualified to judge about the English language and style

Yes

Can be improved

Must be improved

Not applicable

Does the introduction provide sufficient background and include all relevant references?

( )

(x)

( )

( )

Is the research design appropriate?

( )

(x)

( )

( )

Are the methods adequately described?

( )

( )

(x)

( )

Are the results clearly presented?

(x)

( )

( )

( )

Are the conclusions supported by the results?

( )

(x)

( )

( )

Comments and Suggestions for Authors

The authors decribe a case report of a carrier of a novel truncating desmoplakin variant accompanied to a known low-penetrant variant in the NLRP3 gene.

A number of key issues has to be solved.

I recommend that this manuscript will be restructured according to the following suggestions.

We are grateful to the Reviewer for the insightful review and numerous valuable comments. We responded to all questions and modified our manuscript accordingly.

The introduction must be improved by exploring in depth the following issues:

  • the authors only refer to the Padua criteria but in the text they state “She also fulfilled the 2010 Task Force Criteria for the diagnosis of ARVC” (lines 106-107). Therefore, it is necessary to mention and comment the 2010 Task Force Criteria for the diagnosis of ARVC.
  • We added appropriate reference in the introduction.
  • the authors only refer to the ACM overlap with DCM, which is ok, but does not seem to be related to the described case. According to the hypothesis of an “autoinflammatory syndrome predisposition” and the “acute myocarditis-like episodes”, it is essential to discuss the possible overlap between ACM and myocarditis.
  • Thank you for highlighting this. We changed The introduction accordingly (lines 59-62): “Recently, two case reports linked the onset of DSP cardiomyopathy to episodes of myocarditis24, 25. The inflammation of myocardium is often present in patients with ACM and differential diagnosis from myocarditis may be challenging. However, it is not clear whether it is a driving force or only a secondary phenomenon 26. The results of the study by Chelko et al. suggest that activation of nuclear factor-κB (NF-κB), the master regulator of inflammation, immune response, and apoptosis, may be an important mechanism engaged in ACM development.
  • the description of NLRP3 function is unclear and needs to be expanded as well as the impairement caused by the low-penetrant R490K variant during the autoinflammatory syndrome.
  • Thank you for highlighting this. We complemented the introduction  (lines 68-77) and the discussion (lines 276-278, 300-302) accordingly.
  •  
  • “The NLR Family Pyrin Domain Containing 3 protein (NLRP3), or cryopyrin, is an important component of the NLRP3 inflammasome. NLRP3 gain-of-function mutations are the cause of a group of autoinflammatory diseases called cryopyrin-associated periodic syndromes (CAPS)27, typically characterized by recurrent fever, urticarial rash and arthralgia. Unlike these pathogenic mutations, other low-penetrance NLRP3 variants, found also at low frequencies in control populations, can be associated with atypical CAPS with frequent gastrointestinal symptoms28. Inflammation in typical CAPS is driven by cleavage of caspase 1 and release of proinflammatory cytokines interleukin-1β (IL-1β) and interleukin-18 (IL-18), another important effect of inflammasome activation. In contrast, disease symptoms in low-penetrance NLRP3 variant carriers are likely to depend on other mechanisms, possibly through the NF-κB pathway29, 30”.
  •  
  • “Pathogenic heterozygous gain-of-function NLRP3 mutations, located mainly in exon 3, cause excessive release of IL-1β and systemic inflammation, and result in full-blown CAPS.”   “The mode of action of low penetrance NLRP3 variants remains unclear 29, 30. Disease symptoms in these subjects seem independent of the cleavage of caspase 1 and release of proinflammatory cytokines and are likely to depend on other mechanisms 29, 30.”
  •  

The authors state that "NLRP3 mutations cause excessive release of interleukin 1β and systemic inflammation", but no signs of systemic inflammation (normal levels of C-reactive protein) has been found in the proband.

This is consistent with reports on low penetrance NLRP3 variants, where CRP level was normal in 64-67% of symptomatic patients, as in our proband. Also, other inflammatory markers: interleukin-6 and serum amyloid A serum were elevated only in some cases. Therefore, demonstration of systemic inflammation in atypical CAPS is challenging and we do not have results supporting this hypothesis (e.g. concentrations of TNF or S100A12).

We discuss these issues in lines 301-313.

The authors only describe troponin elevation, which may be common in myocarditis but is not specific to myocarditis. I suggest to test interleukin 1β concentration, or that of other inflammatory cytokines in the plasma of the family members.

IL-1β serum measurements do not distinguish even between typical CAPS patients and healthy controls, possibly due to its prompt neutralisation and low and variable  level. Taking into account previously discussed difficulties, demonstration of myocardial inflammation without myocardial biopsy (which is not available) may be impossible.

We added adequate formulations to the study limitations.

The fact that the proband’s brother is carrier of all three identified variants and only has a history of asymptomatic arrhythmias, works against the hypothesis. The authors should better explain their view concerning the differential clinical manifestations and the role of physical exercise as possible trigger. It would be important to test other differences (even not genetic) between the two siblings.

We suppose the asymptomatic phenotype found in the brother may be associated with his young age, incomplete penetrance of the DSP variant and, particularly, the low-penetrant NLRP3 variant – he had no signs of systemic inflammatory disorder. Anyway, Smith et al. found no differences in penetrance of DSP cardiomyopathy attributable to sex or exercise burden.

We hypothesize,  that the differences in severity of disease between the proband and her mildly affected father may be explained by concomitant NLRP3 variant.

We stressed it in the discussion lines 328-332.

Please comment in the discussion the paper: Desmoplakin Cardiomyopathy, a Fibrotic and Inflammatory Form of Cardiomyopathy Distinct From Typical Dilated or Arrhythmogenic Right Ventricular Cardiomyopathy. Circulation 2020;141:1872-1884. It is likely that the inflammatory and episodic nature of the disease of the proband is typical of DSP driven cardiomyopathy.

Thank you for highlighting this. We added an appropriate passage to the discussion (lines 316-320).

“In their recent paper, Smith et al. state that DSP cardiomyopathy is a distinct form of ACM characterized by episodic myocardial injury. Acute episodes occurred in 16/107 (15%) patients, the authors did not mention if they encountered cases with recurrent episodes. In this landmark study however, only DSP variants were evaluated, collected from six tertiary referral centres. There were no differences in penetrance of DSP cardiomyopathy attributable to sex or exercise burden.”

This should be taken into account as a hypothesis alternative to the role of NLRP3 variant.

Please see “4. Study limitations” which we now added.

The method description section is completely missing

We added The methods section to the manuscript and, consequently, additional data on genetic examination in the results (lines 193-195).

Please clarify paragraph 3.3.

We think that discussing the fibrosis pattern in our proband is not essential for the report. We removed the paragraph from the manuscript.

Based on the absence of validation studies it is necessary to provide a study limitations section

Done:

  1. Study limitations.

The patient and her family came to our centre from a remote region of Poland for evaluation in a stable phase of the disease. Our report uses retrospectively collected data which contain no information on cytokine or other inflammatory marker serum levels except for CRP. Neither heart biopsy samples were available for testing. Thus, the role of the low-penetrant NLRP3 variant in the pathogenesis of ACM in our case remains a hypothesis which should be considered with alternative options, i.e. episodic course of ACM attributable specifically to DSP mutations.

Please make sure that genes are in Italic (also in the figures).

We changed the figures accordingly.

Normally DSP+ means the wt allele of DSP, while in fig.5 means presence of the mutation. Please modify the figure to make it easily readable

We modified the figure 5 and the caption.

Please define in Figure 1 legend the dotted line

Defined.

Round 2

Reviewer 1 Report

The authors have addressed each one of the comments/suggestions I raised in my initial review and made appropriate changes/additions to the manuscript. Wherever I suggested any additional experiments - the authors very clearly explained in their response the reasons why this is not possible (lack of myocardial tissue sample from the patient). I believe that the manuscript is significantly improved, is presented in a much more comprehensive way and in its current format merits publication.

Reviewer 2 Report

The authors properly answered to all comments.